# Estimating the Protein Concentration in Rice Grain Using UAV Imagery Together with Agroclimatic Data

**Akira Hama [1],\*** , **Kei Tanaka [2]**, **Atsushi Mochizuki [3]**, **Yasuo Tsuruoka [3]** and **Akihiko Kondoh [4]**

[1] College of Education, Yokohama National University, JSPS Research Fellow, 79-2 Tokiwadai Hodogaya-ku, Yokohama City, Kanagawa 240-8501, Japan

[2] Japan Map Center, 4-9-6 Aobadai, Meguro-Ku, Tokyo 153-8522, Japan; tanaka@jmc.or.jp

[3] Chiba Prefectural Agriculture and Forestry Research Center, 808 Daizenno-Cho, Chiba City, Chiba 266-0062, Japan; a.mchdk5@pref.chiba.lg.jp (A.M.); y.tsrok7@pref.chiba.lg.jp (Y.T.)

[4] Center for Environmental Remote Sensing, Chiba University, 1-33 Yayoi-Cho, Chiba City, Chiba 263-8522, Japan; kondoh@faculty.chiba-u.jp

\* Correspondence: a4k4i8r8a@yahoo.co.jp; Tel.: +81-90-8066-5281

**Abstract:** Global warming and climate change can potentially change not only rice production but also rice nutrient content. To adapt a rice-dependent country's farming to the impacts of climate change, it is necessary to assess and monitor the potential risk that climate change poses to agriculture. The aim of this study was to clarify the relationship between rice grain protein content (GPC) and meteorological variables through unmanned aerial vehicle remote sensing and meteorological measurements. Furthermore, a method for GPC estimation that combines remote sensing data and meteorological variables was proposed. The conclusions of this study were as follows: (1) The accuracy and robustness of the GPC estimation model were improved by evaluating the nitrogen condition with the green normalized difference vegetation index at the heading stage ($GNDVI_{heading}$) and evaluating photosynthesis with the average daily solar radiation during the grain-filling stage ($SR_{grain\text{-}filling}$). GPC estimation considering $SR_{grain\text{-}filling}$ in addition to $GNDVI_{heading}$ was able to estimate the observed GPC under the different conditions. (2) Increased temperature from the transplantation date to the heading stage can affect increased GPC when extreme temperature does not cause the heat stress.

**Keywords:** drone; global warming; nutrient balance; modelling

---

## 1. Introduction

The United Nation's 17 Sustainable Development Goals (SDGs) came into effect in January 2016, and they will continue until 2030. Many SDGs are related to agriculture (e.g., SDG2 Zero hunger: End hunger, achieve food security and improved nutrition and promote sustainable agriculture) that require urgent action from both developing and developed countries [1]. However, implications of climate change, such as global warming, have the potential to interrupt progress being made toward SDGs related to agriculture [2]. With respect to rice production in monsoon Asia, where rapid population growth is driving increased food demand, an increase in temperature and fluctuating precipitation were expected to reduce rice yield, whereas a rise in carbon dioxide ($CO_2$) concentration was expected to increase rice yield because of promoted photosynthesis [3–6]. To adapt agriculture to the impacts of climate change, it is necessary to assess and monitor the potential risk that climate change poses to agriculture. Remote sensing and imagery techniques are useful in monitoring and detecting the effects of climate change on crops [3,7–10]; however, implementation of remote sensing and imagery techniques for monitoring and detecting the effects of climate change is still underdeveloped [11]. Furthermore, while much study has been done on changes in crop production, study on changes in crop nutrient content has been limited [12–14].

Climate change can potentially change not only rice production but also rice grain nutrient content. Rice grain protein content (GPC) is an important source of protein in the highly rice-dependent countries, such as monsoon Asia [13,14]. There is consensus on the impact of increasing $CO_2$ concentration on GPC [13,14]. However, other aspects of climate change, particularly temperature, there are two different hypotheses about the effect of temperature on GPC. For instance, some previous studies have suggested that increasing temperature could reduce GPC [15,16], whereas others have suggested that increasing temperature could increase GPC [17,18]. Thus, there are two conflicting research conclusions about the relationship between temperature and GPC, and this relationship is still unclear. To discuss the effect of climate change on GPC, it is necessary to look at the accumulated studies on changes in GPC due to temperature change.

Furthermore, GPC is also important factor related to the eating quality of cooked rice. Rice with good eating quality is traded as brand rice, and improving the eating quality is seen as a way to enhance farmers' income. Thus, in Japan, there has been considerable basic research and numerous cultivation experiments attempting to control GPC in an appropriate way [16–22]. GPC is related to the firmness and stickiness of cooked rice (e.g., as the GPC decreases, cooked rice becomes sticky). Generally, Japanese prefer sticky cooked rice; therefore, when GPC was low, the eating quality of cooked rice would be highly evaluated [23]. In addition, grain amylose content is also related to the eating quality of cooked rice, it has been revealed that grain amylose content is affected strongly by cultivar and temperature during grain-filling stage [21,22]. On the other hand, GPC is affected strongly by canopy nitrogen content and cultivar [19]. Thus, GPC is affected strongly by nitrogen fertilizer [19]. Among the factors that affect eating quality of cooked rice, GPC is a factor that could be controlled through proper fertilization management [24].

Point sampling and chemical analysis, such as the Kjeldahl method [25], are used generally to measure GPC. If remote sensing can be used to estimate GPC, it would be possible not only to determine the spatial distribution of GPC but also to reduce the labor involved in GPC measurement. Previous studies of GPC estimation via remote sensing have clarified the relationship between GPC and spectral indices [24,26–28]. The findings have shown (1) as the vegetation indices in the grain-filling stage increases, GPC increases; (2) the canopy nitrogen content affects spectral reflectance, and GPC can be estimated indirectly; and (3) regression models for GPC estimation must be remade each year.

The aforementioned studies were based mainly on satellite remote sensing, and only spectral indices were used for GPC estimation. However, approximately 90% of the world's total rice acreage and annual output of rice are concentrated in monsoon Asia, and the growing season for paddy rice includes the rainy season [29]. Consequently, cloud cover has been the principal limitation of optical satellite-based remote sensing. For this reason, remote sensing data have been obtained and analyzed mainly for the dry season. However, remote sensing using the new unmanned aerial vehicle platform (UAV-RS) could be carried out according to the rice growth stage. Hama et al., (2018) examined the optimal observation timing for GPC estimation using UAV-RS [30] and found that to improve the robustness of the regression model, normalized difference vegetation index (NDVI) at the heading stage was the best observation timing for Multi-year GPC estimation. Moreover, as mentioned above, meteorological factors also affect GPC. This indicates that GPC estimation should consider not only vegetation indices but also meteorological factors [31,32].

The aim of this study was to clarify the relationship between GPC and meteorological variables through UAV-RS data together with meteorological measurements. Furthermore, a method for GPC estimation that combines remote sensing data and meteorological variables has been proposed.

## 2. Materials and Methods

### 2.1. Study Sites and UAV Data Acquisition

A UAV-RS data set for three rice cultivars (*Oryza sativa* L. cv. *Koshihikari*, *O. sativa* L. cv. *Fusaotome*, and *O. sativa* L. cv. *Fusakogane*) acquired in Chiba prefectural agriculture and forestry research center was analyzed (Figure 1). *Fusaotme* and *Fusakogane* are allied cultivars. We subdivided test fields into

48 plots with different cultivation conditions (transplantation date, cultivar, and amount of fertilizer). At this test site, the growing seasons differed with change in transplantation date. Table 1 shows the list of cultivation conditions of this study.

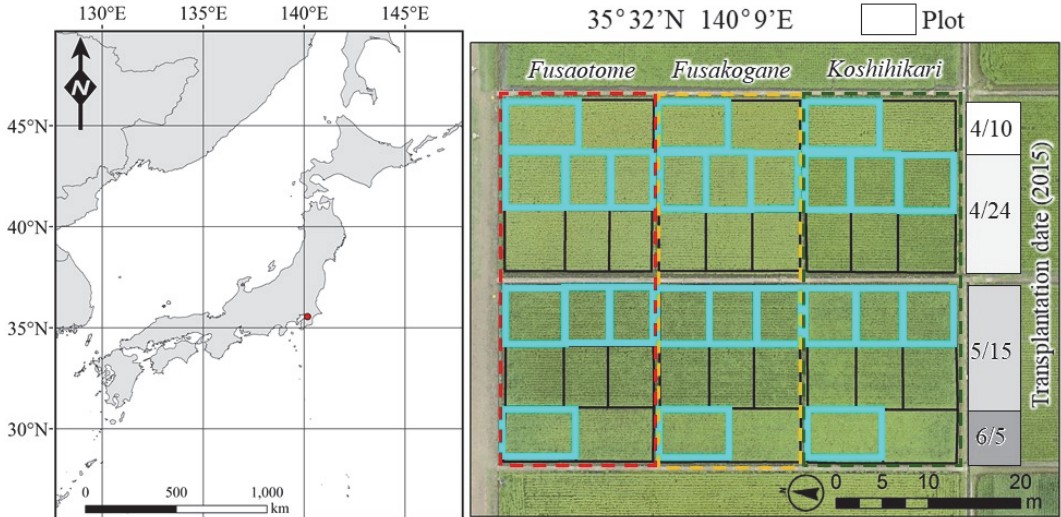

**Figure 1.** Map of the test sites: Chiba Prefectural Agriculture and Forestry Research Center. The grain protein content was observed for each sample (blue plot) from 2015 to 2017.

**Table 1.** The list of cultivation conditions of the test sites (2015 to 2017).

| Year | Test Site | Cultivar | Transplantation Date | Basal Fertilizer gN/m² | Topdressing gN/m² | Growth Stage | |
|------|-----------|----------|----------------------|------------------------|-------------------|--------------|--------|
| | | | | | | Panicle Formation | Heading |
| 2015 | *Chiba* | *Koshihikari* | Apr. 10 | 1.5 | 3.0 | Jun. 10 | Jul. 10 |
| 2015 | *Chiba* | *Koshihikari* | Apr. 24 | 0.0–3.0 | 3.0 | Jun. 21 | Jul. 19 |
| 2015 | *Chiba* | *Koshihikari* | May 15 | 0.0–10.0 | 3.0 | Jun. 29 | Aug. 1 |
| 2015 | *Chiba* | *Koshihikari* | Jun. 5 | 1.5 | 3.0 | Jul. 22 | Aug. 11 |
| 2015 | *Chiba* | *Fusaotome* | Apr. 10 | 3.0 | 3.0 | Jun. 5 | Jul. 5 |
| 2015 | *Chiba* | *Fusaotome* | Apr. 24 | 3.0–9.0 | 3.0 | Jun. 9 | Jul. 12 |
| 2015 | *Chiba* | *Fusaotome* | May 15 | 1.5–7.0 | 3.0 | Jun. 28 | Jul. 21 |
| 2015 | *Chiba* | *Fusaotome* | Jun. 5 | 3.0–4.0 | 3.0 | Jul. 15 | Aug. 7 |
| 2015 | *Chiba* | *Fusakogane* | Apr. 10 | 4.0 | 3.0 | Jun. 5 | Jul. 5 |
| 2015 | *Chiba* | *Fusakogane* | Apr. 24 | 4.0–10.0 | 3.0 | Jun. 10 | Jul. 12 |
| 2015 | *Chiba* | *Fusakogane* | May 15 | 0.0–10.0 | 3.0 | Jun. 29 | Jul. 22 |
| 2015 | *Chiba* | *Fusakogane* | Jun. 5 | 4.0 | 3.0 | Jul. 19 | Aug. 8 |
| 2016 | *Chiba* | *Koshihikari* | Apr. 11 | 2.0 | 3.0 | Jun. 20 | Jul. 15 |
| 2016 | *Chiba* | *Koshihikari* | Apr. 25 | 0.0–2.0 | 3.0 | Jun. 25 | Jul. 24 |
| 2016 | *Chiba* | *Koshihikari* | May 13 | 0.0–2.0 | 3.0 | Jun. 27 | Aug. 5 |
| 2016 | *Chiba* | *Koshihikari* | Jun. 6 | 2.0 | 3.0 | Jul. 24 | Aug. 15 |
| 2016 | *Chiba* | *Fusaotome* | Apr. 11 | 3.0 | 3.0 | Jun. 13 | Jul. 10 |
| 2016 | *Chiba* | *Fusaotome* | Apr. 25 | 3.0–7.0 | 3.0 | Jun. 18 | Jul. 14 |
| 2016 | *Chiba* | *Fusaotome* | May 13 | 3.0–7.0 | 3.0 | Jun. 26 | Jul. 21 |
| 2016 | *Chiba* | *Fusaotome* | Jun. 6 | 3.0 | 3.0 | Jul. 16 | Aug. 7 |
| 2016 | *Chiba* | *Fusakogane* | Apr. 11 | 4.0 | 3.0 | Jun. 13 | Jul. 11 |
| 2016 | *Chiba* | *Fusakogane* | Apr. 25 | 4.0–8.0 | 3.0 | Jun. 19 | Jul. 15 |
| 2016 | *Chiba* | *Fusakogane* | May 13 | 4.0–8.0 | 3.0 | Jun. 27 | Jul. 23 |
| 2016 | *Chiba* | *Fusakogane* | Jun. 6 | 4.0 | 3.0 | Jul. 17 | Aug. 8 |
| 2017 | *Chiba* | *Koshihikari* | Apr. 11 | 2.0 | 2.0 | Jun. 13 | Jul. 12 |
| 2017 | *Chiba* | *Koshihikari* | Apr. 24 | 0.0–2.0 | 2.0 | Jun. 28 | Jul. 20 |
| 2017 | *Chiba* | *Koshihikari* | May 17 | 2.0 | 1.0–2.0 | Jul. 6 | Jul. 31 |
| 2017 | *Chiba* | *Koshihikari* | Jun. 6 | 2.0 | 2.0 | Jul. 20 | Aug. 13 |

**Table 1.** *Cont.*

| Year | Test Site | Cultivar | Transplantation Date | Basal Fertilizer gN/m$^2$ | Topdressing gN/m$^2$ | Growth Stage | |
|------|-----------|----------|----------------------|---------------------------|----------------------|--------------|-----|
| | | | | | | Panicle Formation | Heading |
| 2017 | *Chiba* | *Fusaotome* | Apr. 11 | 3.0 | 3.0 | Jun. 9 | Jul. 7 |
| 2017 | *Chiba* | *Fusaotome* | Apr. 24 | 0.0–5.0 | 3.0 | Jun. 14 | Jul. 11 |
| 2017 | *Chiba* | *Fusaotome* | May 17 | 0.0–3.0 | 1.0–3.0 | Jun. 29 | Jul. 25 |
| 2017 | *Chiba* | *Fusaotome* | Jun. 6 | 3.0 | 3.0 | Jul. 15 | Aug. 6 |
| 2017 | *Chiba* | *Fusakogane* | Apr. 11 | 4.0 | 3.0 | Jun. 10 | Jul. 7 |
| 2017 | *Chiba* | *Fusakogane* | Apr. 24 | 0.0–6.0 | 3.0 | Jun. 15 | Jul. 12 |
| 2017 | *Chiba* | *Fusakogane* | May 17 | 0.0–4.0 | 1.0–3.0 | Jun. 29 | Jul. 25 |
| 2017 | *Chiba* | *Fusakogane* | Jun. 6 | 4.0 | 3.0 | Jul. 17 | Aug. 17 |

The study observation equipment included an electric-powered Multicopter (Zion QC630, enRoute) and a multispectral camera (Yubaflex, BIZWORKS). The Yubaflex is a modified version of the Canon PowerShot S100 camera, which takes images at the green, red, and near-infrared (NIR) bands. The bandwidth of each band was as follows: Green 500–600nm, Red 600–850nm, and NIR 700–1050nm (wavelengths showing the maximum spectral response of each band: Green 550 nm, Red 600 nm, and NIR 850 nm). The image was made up of 12 million pixels (4000 × 3000). The camera can also convert the observed digital number to radiance using the dedicated software Yubaflex 3.1 [33]. In this study, we used the Yubaflex for vegetation index monitoring and GPC estimation.

The UAV-based observations were acquired once a week between 10:00 and 10:30 a.m. local time, under both clear and cloudy sky conditions. The flight altitude was 50 m above ground level (ground resolution: 1.8 cm), and the overlap of each image was 70%. The settings of the Yubaflex were shutter speed priority mode, shutter speed set to 1/1000 sec, ISO sensitivity set to Auto, and interval of shooting set to the minimum value (approximately 4 sec).

*2.2. Image Processing and Analysis*

The images taken with the Yubaflex, after being converted to radiance using the dedicated software Yubaflex3.1, were used to create orthomosaic images using Structure from Motion–Multi View Stereo (SfM–MVS) software (Agisoft PhotoScan Professional v1.4.1).

Yubaflex-based NDVI values are relatively lower than the other multispectral camera-based NDVI [34]. The bandwidth of Yubaflex red band is 600–850nm, and some of NIR wavelength are included. Therefore, Yubaflex red band continue to respond to high aboveground biomass, and NDVI values become relatively low. In this study, we used the green NDVI (GNDVI) [35] in order not to become vegetation index values low. GNDVI was calculated based on equation (1):

$$\text{GNDVI} = (\text{NIR}_{\text{Yubaflex}} - \text{Green}_{\text{Yubaflex}}) / (\text{NIR}_{\text{Yubaflex}} + \text{Green}_{\text{Yubaflex}}), \tag{1}$$

where GNDVI is the Yubaflex-based GNDVI, and $\text{NIR}_{\text{Yubaflex}}$ and $\text{Green}_{\text{Yubaflex}}$ are the Yubaflex-based NIR band and Green band radiances, respectively. The average GNDVI was then calculated for each plot (Figure 1) using ESRI ArcGIS 10.4.

In general, the vegetation indices decreased with decreasing solar zenith angle. This response was affected significantly by the growth stage and diffuse/direct light conditions [36]. Ishihara et al. (2015) [36] reported that the decreasing response of the vegetation indices to the decreasing solar zenith angle was high during the middle growth stage and low at the heading stage. In addition, the response of vegetation indices to the solar zenith angle was evident under clear sky conditions at large solar zenith angles (less than 20°) but almost negligible under cloudy sky conditions [36]. In this study, UAV-based observations were acquired between 10:00 and 10:30 a.m. local time. The solar zenith angle at the observation was more than 20°. The sunlight conditions appear to have a low effect on the consistency of GNDVI in this study.

### 2.3. Analysis of Collected Samples and Meteorological Data

At the Chiba site, GPC (converted to moisture content 15%) was observed from 2015 to 2017. Samples in which lodging occurred were excluded, and the analysis covered 24 samples from 2015, 23 samples from 2016, and 21 samples from 2017. Total nitrogen of grain adjusted with a grain thickness of 1.8 mm was observed using the NC analyzer (Sumigraph NC-900, Sumika Analysis Service [37]). Subsequently, the conversion factor (5.95) of nitrogen–protein was multiplied to covert total nitrogen to GPC.

In this study, the daily mean solar radiation and the daily mean temperature from agricultural weather data obtained from 1 km grid squares [38] were analyzed for GPC estimation. These agro-meteorological grid square data are based on data from the ground observation station, known as the Automated Meteorological Data Acquisition System network.

### 2.4. GPC Estimation

Two types models (simple linear regression model and multiple regression model) for the estimation of GPC were developed; simple linear regression (SLR) model was built by using $GNDVI$ at the heading stage ($GNDVI_{heading}$) as explanatory variable, while multiple regression (MR) model was built by using $GNDVI_{heading}$ and average daily solar radiation during the grain-filling stage ($SR_{grain-filling}$) as explanatory variables. Figure 2 denotes the GPC estimation process.

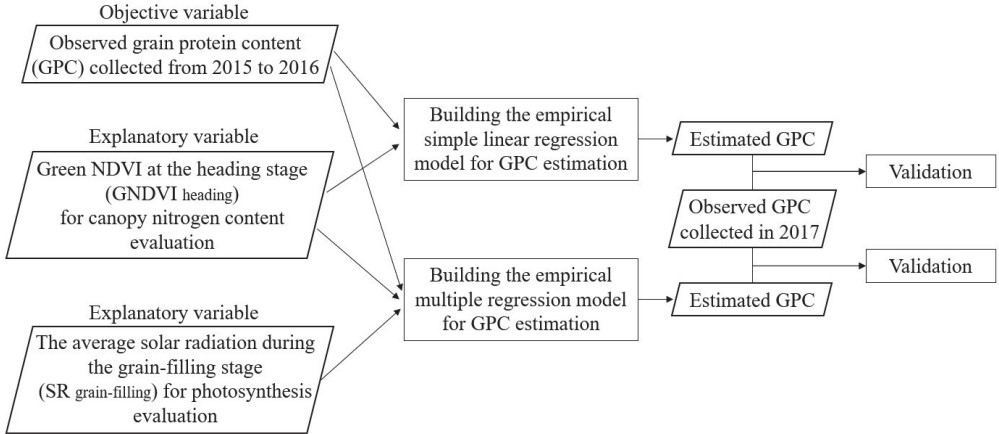

**Figure 2.** The process of grain protein content (GPC) estimation.

The samples collected from 2015 to 2016 were used to build the empirical GPC estimation models for three cultivars (*Koshihikari*, *Fusaotome*, and *Fusakogane*). At first, a SLR analysis for three cultivars was conducted to explore the relationship between GPC and $GNDVI_{heading}$. Following the SLR analysis, the GPC estimation models were obtained as:

$$GPC = m \, GNDVI_{heading} + k, \tag{2}$$

where GPC is the grain protein content (%), $GNDVI_{heading}$ is the Yubaflex-based GNDVI at the heading stage, $m$ is the coefficient, and $k$ is the intercept. Then, a MR analysis for three cultivars was conducted to explore the relationship between GPC, $GNDVI_{heading}$, and $SR_{grain-filling}$. Following the multiple regression analysis, the GPC estimation models were obtained as:

$$GPC = m_1 \, GNDVI_{heading} + m_2 \, SR_{grain-filling} + k, \tag{3}$$

where GPC is the grain protein content (%), $GNDVI_{heading}$ is the Yubaflex-based GNDVI at the heading stage, $SR_{grain-filling}$ is the average daily solar radiation during the grain-filling stage ($MJ/m^2$), $m_i$ are the coefficients, and $k$ is the intercept. $GNDVI_{heading}$ was used for evaluation of canopy nitrogen content, and $SR_{grain-filling}$ was used for evaluation of photosynthesis. The results of both single and multiple

regression analysis were evaluated by the coefficient of determination ($R^2$), the *p* value. A *p* value lower than 0.05 was considered statistically significant. Finally, GPC estimation model robustness was validated by using the observed GPC collected in 2017. The validation results were evaluated in terms of the root mean square error (RMSE).

A sensitivity analysis of the MR model was performed in terms of the rate of change in the output value resulting from a change of each input parameter while keeping all other parameters constant [32]. The maximum, minimum, and average values collected in the experiment from 2015 to 2017 were used for sensitivity analysis.

Hama et al. (2018) used temperature instead of $SR_{grain-filling}$ for GPC estimation [30]. In the case of *Koshihikari*, the average temperature from 0 to 20 days after the heading stage was used for the GPC estimation [30]. In the cases of *Fusakogane* and *Fusaotome*, the average temperatures from 0 to 30 days after heading stage were used for the GPC estimation. In this study, the determination of the $SR_{grain-filling}$ duration was based on Hama et al. (2018) [30]. In the case of *Koshihikari*, the average daily solar radiation from 0 to 20 days after the heading stage was used as $SR_{grain-filling}$. In the cases of *Fusakogane* and *Fusaotome*, the average daily solar radiation from 0 to 30 days after the heading stage was used as $SR_{grain-filling}$.

## 3. Results

### 3.1. Regression Analysis for GPC Estimation

Table 2 denotes the results of the SLR analysis. The regression coefficient (*m*), intercept (*k*), $R^2$, and *p* value are listed in Table 2. The results of SLR analysis demonstrate that the higher the $GNDVI_{heading}$, the higher the GPC. The *p* values of $GNDVI_{heading}$ were lower than 0.05, and these were considered statistically significant.

**Table 2.** The results of simple linear regression analysis for rice grain protein content estimation. *n* = number of samples, *m* = coefficient, *k* = intercept, $GNDVI_{heading}$ = Yubaflex-based green normalized difference vegetation index at the heading stage.

| Cultivar | *n* | Coefficient | *k* | *p* Value | $R^2$ |
|---|---|---|---|---|---|
| | | *m* | | $GNDVI_{heading}$ | |
| *Koshihikari* | 15 | + 10.82 | + 0.79 | $3.4 \times 10^3$ ** | 0.495 |
| *Fusaotome* | 16 | + 9.73 | + 1.53 | $5.9 \times 10^4$ *** | 0.582 |
| *Fusakogane* | 16 | + 8.38 | + 2.35 | $4.8 \times 10^4$ *** | 0.593 |

*: *p* value < 0.05, **: *p* value < 0.01, ***: *p* value < 0.001.

Table 3 denotes the results of the MR analysis. The regression coefficients ($m_i$), intercept (*k*), $R^2$, and *p* value are listed in Table 3. The results of MR analysis demonstrate that the higher the $GNDVI_{heading}$, the higher the GPC, and the higher the $SR_{grain-filling}$, the lower the GPC. Furthermore, with respect to the coefficients and intercepts, *Fusaotome* and *Fusakogane* were similar, but *Koshihikari* differed from the other two cultivars. The *p* values of $GNDVI_{heading}$ and $SR_{grain-filling}$ were lower than 0.05, and these were considered statistically significant.

**Table 3.** The results of multiple regression analysis for rice grain protein content estimation. *n* = number of samples, $m_i$ = coefficient, *k* = intercept, $GNDVI_{heading}$ = Yubaflex-based green normalized difference vegetation index at the heading stage, $SR_{grain-filling}$ = average daily solar radiation during the grain-filling stage ($MJ/m^2$).

| Cultivar | *n* | Coefficient | | *k* | *p* Value | | $R^2$ |
|---|---|---|---|---|---|---|---|
| | | $m_1$ | $m_2$ | | $GNDVI_{heading}$ | $SR_{grain-filling}$ | |
| ***Koshihikari*** | **15** | **+ 9.93** | − 0.08 | + 2.74 | $3.1 \times 10^3$ ** | $3.1 \times 10^2$ * | 0.568 |
| *Fusaotome* | 16 | + 9.21 | − 0.12 | + 4.13 | $6.8 \times 10^5$ *** | $2.6 \times 10^4$ *** | 0.796 |
| *Fusakogane* | 16 | + 8.98 | − 0.12 | + 4.09 | $2.5 \times 10^5$ *** | $3.8 \times 10^4$ *** | 0.712 |

*: *p* value < 0.05, **: *p* value < 0.01, ***: *p* value < 0.001.

Table 4 shows the results of sensitivity analysis of MR models. The GPC variation (%) in GNDVI$_{heading}$ of *Koshihikari*, *Fusaotome*, and *Fusakogane* were –7.7 to +11.2, –16.4 to +10.2, and –12.7 to +11.8, respectively. The GPC variation (%) in SR$_{grain-filling}$ of *Koshihikari*, *Fusaotome*, and *Fusakogane* were –6.3 to +6.6, –6.7 to +7.9, and –6.8 to +8.7, respectively. In all cultivars, sensitivity analysis revealed a higher GNDVI$_{heading}$ sensitivity to GPC. In other words, GNDVI$_{heading}$ had a greater effect on GPC.

**Table 4.** Sensitivity analysis for multiple regression model. Avg = average value collected in the experiment from 2015 to 2017, Min = minimum value collected in the experiment from 2015 to 2017, Max = maximum value collected in the experiment from 2015 to 2017, grain protein content (GPC) variation = rate of change in the estimated GPC value compared with the average value (%), GNDVI$_{heading}$ = Yubaflex-based green normalized difference vegetation index at the heading stage, SR$_{grain-filling}$ = average daily solar radiation during the grain-filling stage (MJ/m$^2$).

| Cultivar | Variable | Avg | Min | Max | GPC Variation % (min Value) | GPC Variation % (max Value) |
|---|---|---|---|---|---|---|
| *Koshihikari* | GNDVI$_{heading}$ | 0.572 | 0.518 | 0.651 | – 7.7 | + 11.2 |
| *Koshihikari* | SR$_{grain-filling}$ | 17.64 | 11.87 | 23.20 | + 6.6 | – 6.3 |
| *Fusaotome* | GNDVI$_{heading}$ | 0.594 | 0.462 | 0.676 | – 16.4 | + 10.2 |
| *Fusaotome* | SR$_{grain-filling}$ | 18.08 | 13.19 | 22.23 | + 7.9 | – 6.7 |
| *Fusakogane* | GNDVI$_{heading}$ | 0.603 | 0.499 | 0.699 | – 12.7 | + 11.8 |
| *Fusakogane* | SR$_{grain-filling}$ | 18.08 | 12.78 | 22.23 | + 8.7 | – 6.8 |

The comparison between the estimated GPC and the observed GPC is shown in Figure 3. With respect to SLR-based estimation, the RMSE of *Koshihikari* was 0.61 (average observed GPC 7.08%), the RMSE of *Fusaotome* was 0.67 (average observed GPC 7.36%), and the RMSE of *Fusakogane* was 0.58 (average observed GPC 7.47%). With respect to MR-based estimation, the RMSE of *Koshihikari*, *Fusaotome*, and *Fusakogane* were 0.42, 0.34, and 0.33, respectively. The results of MR-based estimation outperformed the SLR-based estimation.

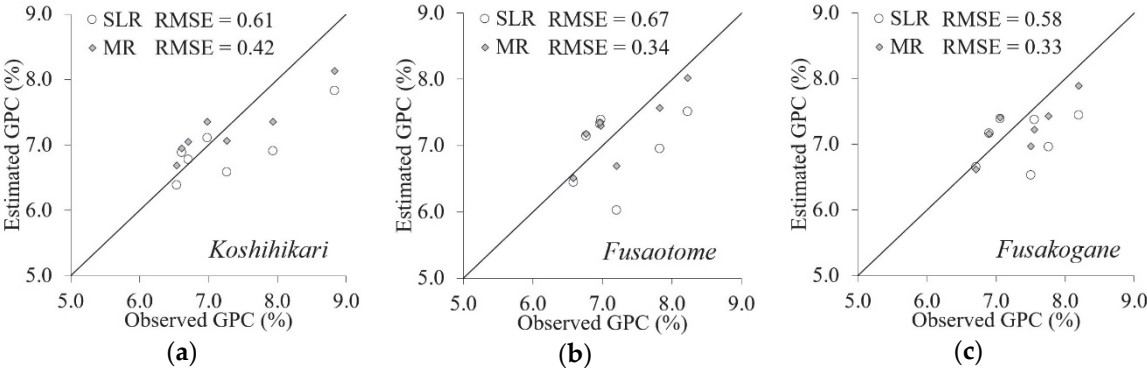

**Figure 3.** Comparison between estimated and observed rice grain protein content (GPC) in 2017: (**a**) shows the estimated GPC of *Koshihikari*; (**b**) shows the estimated GPC of *Fusaotome*; and (**c**) shows the estimated GPC of *Fusakogane.* Circles show simple linear regression (SLR) based results, and diamonds show multiple regression (MR) based results.

*3.2. GNDVI Time Series*

Figure 4 shows the GNDVI time series in *Koshihikari*, *Fusaotome*, and *Fusakogane* transplanted on May 13 2016. In all cultivars, GNDVI increased toward the heading stage and declined gradually after the heading stage. The peak of GNDVI in *Koshihikari* was 82 days after transplantation, whereas it was 68 days after transplantation for *Fusaotome* and *Fusakogane*. The time series patterns of the allied cultivars (*Fusaotome* and *Fusakogane*) were similar, and the peak of GNDVI was 14 days earlier than

with *Koshihikari*. GNDVI time series clearly showed change speed with development depending on the cultivar. In addition, the peak of GNDVI was recorded at the heading stage in all cultivars.

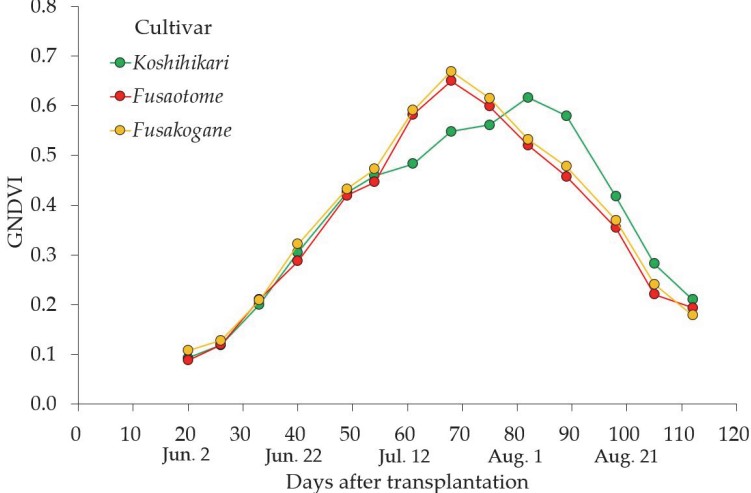

**Figure 4.** Time series of green normalized vegetation index (GNDVI) in the test site (2016). Green circles and the green line are *Koshihikari*, red circles and the red line are *Fusaotome*, and orange circles and the orange line are *Fusakogane*.

Figure 5 shows the GNDVI time series in *Koshihikari* plot of 2016 with different transplantation dates. The days after transplantation at the peak of GNDVI of the transplanting dates April 11, April 25, May 12, and Jun 3 were 100 days, 86 days, 83 days and 68days, respectively. The days after transplantation at the peak of GNDVI decreased when the transplantation date was later. The maximum value of GNDVI of the transplantation dates April 11, April 25, May 12, and Jun 3 were 0.586, 0.622, 0.631, and 0.675, respectively. Although the amount of fertilizer was the same (basal fertilizer 2.0gN/m$^2$, topdressing 3.0gN/m$^2$), the maximum value of GNDVI increased when the transplantation date was later.

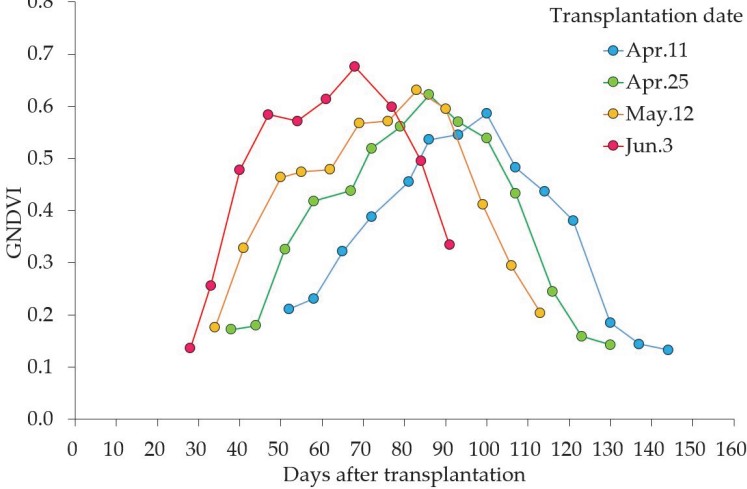

**Figure 5.** Time series of green normalized difference vegetation index (GNDVI) of *Koshihikari* with different transplantation dates (2016). The cultivar and the amount of fertilizer are the same. The blue line denotes the time series of GNDVI of the transplantation date April 11, the green line shows the time series of GNDVI of the transplantation date April 25, the orange line shows the time series of GNDVI of the transplantation date May 12, and the red line shows the time series of GNDVI of the transplantation date Jun 3.

Figure 6 shows the correlation between the average temperature from the transplantation date to the heading stage and the GNDVI$_{heading}$. There were four plots with the same amount of fertilizer in a year for each cultivar. The $R^2$ of *Koshihikari* in 2015, 2016, and 2017 were 0.948, 0.951, and 0.868, respectively. The $R^2$ of *Fusaotome* in 2015, 2016, and 2017 were 0.981, 0.949, and 0.956, respectively. The $R^2$ of *Fusakogane* in 2015, 2016, and 2017 were 0.989, 0.904, and 0.946, respectively. Although the same amount of fertilizer was used for the same cultivars, GNDVI$_{heading}$ increased as the average temperature from the transplantation date to the heading stage increased. The *p* values of *Koshihikari* in 2015, 2016, and 2017 were 0.023, 0.024, and 0.068, respectively. The *p* values of *Fusaotome* in 2015, 2016, and 2017 were 0.009, 0.026, and 0.022, respectively. The *p* values of *Fusakogane* in 2015, 2016, and 2017 were 0.006, 0.048, and 0.027, respectively. The *p* values were lower than 0.05 except for *Koshihikari* in 2017, and these were considered statistically significant. Although the cultivar was the same, the correlation between the average temperature from the transplantation date to the heading stage and the GNDVI$_{heading}$ was varied with respect to collected year.

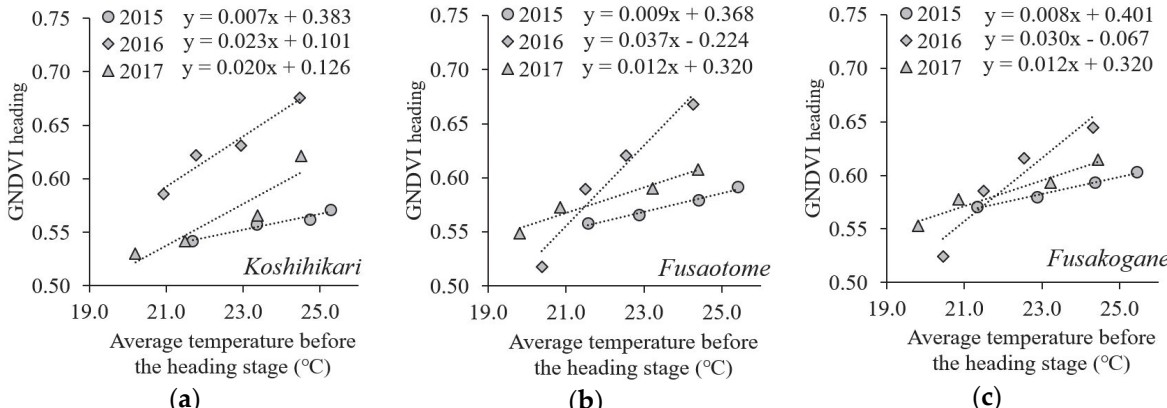

**Figure 6.** Correlation between average temperature from the transplantation date to the heading stage (°C) and green normalized difference vegetation index (GNDVI) at the heading stage: (**a**) shows the results of *Koshihikari*; (**b**) shows the results of *Fusaotome*; and (**c**) shows the results of *Fusakogane*. Circles show the samples in 2015, diamonds show the sample in 2016, and triangles show the sample in 2017.

Figure 7 shows the spatial and temporal variability in GNDVI. The value of GNDVI varied in plots with different cultivation conditions (transplantation date, cultivar, amount of fertilizer). Even if the transplantation date and cultivar were the same, the value of GNDVI increased as the amount of fertilizer increased.

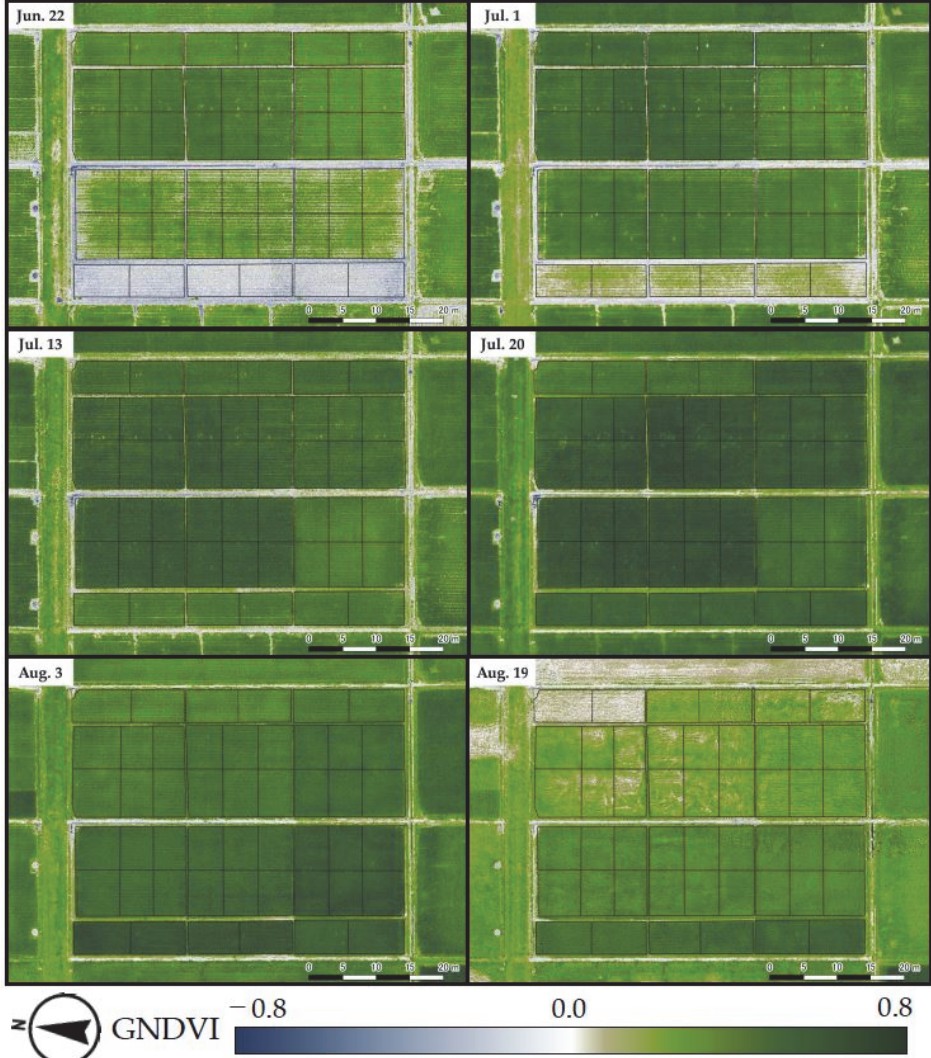

**Figure 7.** Spatial and temporal variability in green normalized difference vegetation index (GNDVI) from unmanned aerial vehicle imagery in 2016. Black line is the boundary of plot.

## 4. Discussion

The regression coefficient both SLR and MR analysis showed that there was an increase in GPC with increasing $GNDVI_{heading}$. The *p* values of $GNDVI_{heading}$ were significant at 5% or less in all cultivars. In the two types (SLR and MR) models for GPC estimation, $GNDVI_{heading}$ was used for evaluation of canopy nitrogen content. The relation between $GNDVI_{heading}$ and canopy nitrogen content is supported by the findings of Inoue et al. (2012) who examined simple and robust methods for remote sensing of canopy nitrogen content [39].

In the MR models for GPC estimation, $SR_{grain-filling}$ was used for evaluation of photosynthesis. There was a decline in GPC with increasing $SR_{grain-filling}$. In addition, the *p* values of $SR_{grain-filling}$ were significant at 5% or less in all cultivars. These findings were consistent with those of a previous study in which GPC was shown to decrease because of the increased carbohydrate production via photosynthesis in rice grains [13,14]. In addition, the $R^2$ and RMSE of the MR models including the $SR_{grain-filling}$ outperformed the previous study's method (SLR models) for GPC estimation, in which only the correlation between the $GNDVI_{heading}$ and the observed GPC was used (Figure 3). GPC estimation that considers only $SR_{grain-filling}$ in addition to $GNDVI_{heading}$ could improve estimation accuracy and improve the robustness of the estimation model. Furthermore, the observed GPC under the different conditions could be estimated with high accuracy without remaking the regression models

for GPC estimation for each transplantation date. This was also an improvement on the previous studies of GPC estimation using remote sensing.

Sensitivity analysis revealed that $GNDVI_{heading}$ had a greater effect on GPC (Table 4). As previously stated, there is a correlation between GNDVI and canopy nitrogen content [39], these findings were consistent with those of a previous study in which nitrogen fertilizer strongly affected GPC [18,23].

It is worth noting that the parameters of the GPC estimation model are specific to the cultivar. The characteristics of each cultivar, such as growth speed and the nutrient translocation from source to sink, are different. According to previous studies, the effect of cultivar difference is as great as that of fertilization [19]. It is necessary to obtain parameters specific to the cultivar based on MR analysis using $GNDVI_{heading}$ and $SR_{grain-filling}$; therefore, it is necessary to perform tests with multiple transplantation dates, as done in this study, to optimize the parameters of the GPC estimation model. However, the parameters of the GPC estimation model might be similar in cases of allied cultivars, such as the *Fusaotome* and *Fusakogane* cultivars in this study.

MR models could estimate the observed GPC under the different conditions as described above; however, MR models would underestimate the GPC in case topdressing after the heading stage. The timing of topdressing affects protein, and topdressing after the heading stage greatly increases GPC [40–42]. MR models use GNDVI at the heading stage; therefore, the effect of topdressing after the heading stage cannot be considered.

As shown in Figure 4, the peak of GNDVI was recorded at the heading stage in all cultivars. This finding was consistent with those of a previous study in which monitoring the paddy field using remote sensing [43–45]. GNDVI time series (Figure 4) showed change speed with development depending on the cultivar.

The days after transplantation at the peak of GNDVI decreased when the transplantation date was later (Figure 5). Changes in GNDVI time series due to differences in transplantation date are in agreement with the findings of the previous studies that the development speed of paddy rice becomes faster when the temperature increased during the growing season [46,47].

As shown in Figure 6, $GNDVI_{heading}$ increased when the average temperature from the transplantation date to the heading stage increased. In addition, the maximum value of GNDVI increased when the transplantation date was later (Figure 5). Nitrogen absorbed by plants is decomposed into inorganic nitrogen in the soil. Previous study has demonstrated that the amount of inorganic nitrogen increases as the temperature rises [48]. Therefore, this study might show that as the average temperature from the transplantation date to the heading stage increased, the amount of inorganic nitrogen and nitrogen absorbed by the plants increased, even when the amount of fertilizer used was unchanged. Consequently, $GNDVI_{heading}$ increased as the transplantation date became later.

There are two different hypotheses concerning the effect of temperature on GPC: some studies suggest that increasing temperature can reduce GPC [15,16], whereas others suggest that increasing temperature can increase GPC [17,18]. As described above, the findings of the current study show that increasing temperature from the transplantation date to the heading stage can increase GPC. However, the findings of current study only follow the increase of temperature from spring to summer. Furthermore, the extreme temperature or temperature well above the average during crop development could cause the heat stress and crops can show rapid development without GPC increasing [32]. Therefore, the findings of current study suggested that increasing temperature from the transplantation date to the heading stage can increase GPC when extreme temperature does not cause the heat stress.

## 5. Conclusions

We used UAV-RS data and meteorological measurements to clarify the relationship between GPC and meteorological variables. Furthermore, a method for GPC estimation that combines remote sensing data and meteorological variables has been proposed.

We proposed the simple method for GPC estimation by using $GNDVI_{heading}$ together with $SR_{grain-filling}$. $GNDVI_{heading}$ was used for evaluation of canopy nitrogen content, and $SR_{grain-filling}$

was used for evaluation of photosynthesis. MR analysis and the GPC estimation models for three cultivars showed that the higher the GNDVI$_{heading}$, the higher the GPC, and the higher the SR$_{grain-filling}$, the lower the GPC. Additionally, the *p* value of GNDVI$_{heading}$ and SR$_{grain-filling}$ were significant at 5% or less in all cultivars. The validation of GPC estimation showed that the RMSE of *Koshihikari* was 0.42 (average observed GPC 7.08%), the RMSE of *Fusaotome* was 0.34 (average observed GPC 7.36%), and the RMSE of *Fusakogane* was 0.33 (average observed GPC 7.47%). Although the parameters of the GPC estimation model are specific to the cultivar, it was possible to improve GPC estimation accuracy and model robustness. Estimation of GPC that only considers SR$_{grain-filling}$ in addition to GNDVI$_{heading}$ could estimate the observed GPC under the different conditions without remaking the regression models for each transplanting date.

The GNDVI time series and the correlation between the average temperature from the transplantation date to the heading stage and the GNDVI$_{heading}$ showed that GNDVI$_{heading}$ increased when the temperature from the transplantation date to the heading stage increased although the amount of fertilizer was the same. In addition, MR models for GPC estimation showed that GPC increased when GNDVI$_{heading}$ increased. Therefore, this study has shown that increasing temperature from the transplantation date to the heading stage can affect increased GPC when extreme temperature does not cause the heat stress.

**Author Contributions:** Conceptualization, A.H.; methodology, A.H.; software, A.H.; validation, A.H. and A.M.; formal analysis, A.H.; investigation, A.H., K.T., A.M., Y.T. and A.K.; resources, A.H., K.T., A.M., Y.T. and A.K.; data curation, A.H. and A.M.; writing—original draft preparation, A.H.; writing—review and editing, A.K.; supervision, A.K.; project administration, A.H.; funding acquisition, A.H. and A.K. All authors have read and agreed to the published version of the manuscript.

**Funding:** This work was supported by JSPS KAKENHI Grant Number 17J01308, 19J00437.

**Acknowledgments:** The authors express thanks to the member of Chiba prefectural agriculture and forestry research center for their helpful support in the field observation.

**Conflicts of Interest:** The authors declare no conflict of interest.

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
