# Peer review of "Estimating the Protein Concentration in Rice Grain Using UAV Imagery Together with Agroclimatic Data"

_agronomy, doi:10.3390/agronomy10030431_

Round 1

Reviewer 1 Report

Dear authors and editor, thank you for giving me the opportunity to review this interesting paper.

The application of remote sensing techniques to investigate the effects of climate change on food grains is becoming increasingly interesting for the scientific community and the farmers.

The approach of this study is simple but effective. However, I believe the manuscript would benefit from a major review and English editing before being considered for the publication in this Journal.

Probably, too few references are considered in the Introduction section. As an example, some recent papers on the impact of climate change on rice production are suggested at the end of comments.

The statistical approach needs to be clarified in some parts. The results don’t show any connection with the increase of CO2. Furthermore, the results highlight some lack related to the choice of vegetation index.

Here are some specific comments:

Line 26: I suggest deleting “This study showed that”

Lines 35-36: the sentence “However, climate change, such as global warming” doesn’t sound correct to me, since global warming is one of the implications of CC, I think it would be more appropriate to state. “However, the implications of climate change, such as…”

Lines 37-40: the word “increase” is repeated 5 times in only one sentence

Line 41: “monitoring” should be “monitor”

Line 49: this statement needs a reference

Line 69: check for extra capital letter

Line 83: “considering” should be “consider”

Line 85: check verbs

Line 90: UAV remote sensing should be UAV-RS

Line 126: please add a reference for NC analyser

Lines 166-167 and Figures 2 and 3: I might be wrong, but aren’t the NDVI values too low? It looks like 0.366 is the maximum value reached in the panicle formation stage. Having a look to previous literature I found out higher values for this stage (i.e.:

  1. González-Betancourt, M.; Mayorga-Ruíz, Z.L. Normalized difference vegetation index for rice management in el Espinal, Colombia. DYNA 2018, 85, 47–56.
  2. Li, P.; Jiang, L.; Feng, Z.; Sheldon, S.; Xiao, X. Mapping rice cropping systems using Landsat-derived Renormalized Index of Normalized Difference Vegetation Index (RNDVI) in the Poyang Lake Region, China. Front. Earth Sci. 2016, 10, 303–314.)

I saw you included in the discussion the reasons for these values, but my question is: wouldn't have been more appropriate to adopt a different vegetation index, not using the NIR? Why did you choose NDVI? I think the reason should be included and explained in your Method or Discussion section.

Fig. 4: Was correlation analysis conducted with aggregated data? In this case, did a dataset of 4 values provide a significant correlation? Could you provide p-values?

Lines 271-272 and 298-299: Did this study investigated the effects of CO2 on GPC as stated in this sentence? I think something is missing.

Lines 294-295: this statement has already been presented in the Discussion section and there is not need to repeat it here.

Suggested references:

Castells-Quintana, D.; Lopez-Uribe, M.d.; McDermott, T.K.J. Geography, institutions and development: A review of the long-run impacts of climate change. Clim. Dev. 2017, 9, 452–470.

Cogato, Alessia, Franco Meggio, Massimiliano De Antoni Migliorati, and Francesco Marinello. “Extreme Weather Events in Agriculture: A Systematic Review.” Sustainability (Switzerland) 11, no. 9 (2019): 1–18.

Singh, K.K.; Kalra, N. Simulating impact of climatic variability and extreme climatic events on crop production. Mausam 2016, 67, 113–130.

Reviewer 2 Report

The topic of the manuscript entitled “Estimating the Protein Concentration in Rice Grain using UAV Imagery together with Agroclimatic Data” is interesting and falls into scope of the journal.

Some improvements in the manuscript are required.

Introduction should be focused on the topic of the manuscript. The first paragraph of the manuscript is about CO2. I do not think it is connected with the aim of the study.

There is no detailed information about experimental design. What type of the design was applied. It was randomized complete block design or other type of design? Factors of the design should be clearly presented.

Authors use word “compartment” for each experimental unit. I suggest to use rather “plot” instead of “compartment”.

Values of NDVI were quite low. During intensive crop growth NDVI is usually much greater than 0.5. Could you explain why these values were much lower? Is the UAV based NDVI similar to satellite NDVI for example from Sentinel-2? Authors wrote in the Discussion: “Yubaflex-based NDVI values are relatively lower than the other multispectral camera-based NDVI. … “…NDVI values from both
225 systems had a high correlation (R2 = 0.968)”. Could you add the figure which presents relationships between these two different NDVI?

The regression analyses, both simple and multiple are based on very small number of observations (n=4 or n=7). What is the each value used for the analyses. It was value from one plot or averaged data? It should be clearly stated. It is not clear form which year or years the data were used.

Titles of the figures and the tables should be self-explanatory, i.e. clear enough without reading all the manuscript. Current titles are not clear.

How decrease of NDVI can be explained in early stages? For example in Fig. 2 at about 50 days after transplantation substantial decrease of NDVI was observed for June 3. What was the reason?

Round 2

Reviewer 1 Report

Based on my previous review, I believe the quality of the manuscript has been improved. The authors improved the introduction, changed the vegetation index and adjusted the statistics. I have only one remark:

Line 232: “expect” should be “except".

I think the quality of some images could be improved, but this will be the Editor's decision.

Thank you for giving me the opportunity of reviewing this manuscript.

Reviewer 2 Report

The manuscript was improved according all my comments.

I propose only one minor change, i.e. equations of multiple regression are presented separately to other results of multiple regression. In my opinion the equations (2, 3, and 4) should be a part of the results presented in Table 2. It would more clear for readers to understand main results of the study.
